# Understanding Housing Prices Using Geographic Big Data: A Case Study in Shenzhen

**Xufeng Jiang [1], Zelu Jia [2,\*], Lefei Li [3,4] and Tianhong Zhao [3]**

[1] Aerospace Information Research Institute, Chinese Academy of Sciences, Beijing 100094, China; jiangxf@radi.ac.cn
[2] Government Services Data Bureau of Bao'an District Shenzhen Municipality, Shenzhen 518000, China
[3] School of Architecture and Urban Planning, Shenzhen University, Shenzhen 518000, China; lilefei@didiglobal.com (L.L.); zhaotianhong2016@email.szu.edu.cn (T.Z.)
[4] DiDi Chuxing, Beijing 100085, China
[\*] Correspondence: jiazelu@baoan.gov.cn

**Abstract:** Understanding the spatial pattern of urban house prices and its association with the built environment is of great significance to housing policymaking and urban planning. However, many studies on the influencing factors of urban housing prices conduct qualitative analyses using statistical data and manual survey data. In addition, traditional housing price models are mostly linear models that cannot explain the distribution of housing prices in urban areas. In this paper, we propose using geographic big data and zonal nonlinear feature machine learning models to understand housing prices. First, the housing price influencing factor system is built based on the hedonic pricing model and geographic big data, and it includes commercial development, transportation, infrastructure, location, education, environment, and residents' consumption level. Second, a spatial exploratory analysis framework for house price data was constructed using Moran's I tools and geographic detectors. Finally, the XGBoost model is developed to assess the importance of the variables influencing housing prices, and the zonal nonlinear feature model is built to predict housing prices based on spatial exploration results. Taking Shenzhen as an example, this paper explored the distribution law of housing prices, analyzed the influencing factors of housing prices, and compared the different housing price models. The results show that the zonal nonlinear feature model has higher accuracy than the linear model and the global model.

**Keywords:** housing price model; machine learning; big data; regression analysis; built environment

## 1. Introduction

With the intensification of urbanization, the demand for house leasing and house purchasing has also continued to increase, and the housing price problem is related not only to people's living standards but also to the development of the national economy [1]. The gap between housing prices and house value [2], the relationship between the specific layout of the housing price in a city and its nearby location and environmental factors [3], has become an urgent problem for government departments and a universal concern of individuals. Therefore, revealing the law of housing prices has positive significance for urban planning and real estate pricing [4].

In recent years, researchers from economics, urban planning, geography, politics, and computer science have conducted considerable amounts of research on housing prices to understand the factors that influence housing prices [5–7]. Housing prices in a city have a close relationship with the spatial location and surrounding settings. The price of houses varies in different locations. W. Alonso, an American economist, proposed the bid rent function in 1964 [8]. Based on this function and the single-center hypothesis, he concluded that urban housing rent is a negative correlation function that decreases with distance from the center of the city to the periphery. However, due to the expansion of

the scale of cities and the improvement in traffic conditions, the urban system is diverse and complex. It is difficult to explain urban housing prices using the distance to the city center as a model to evaluate the distribution of housing prices. Therefore, it is necessary to construct a characteristic system from multiple factors, such as society, economy, location, and environment, to make the housing price model more accurate and reasonable.

In terms of modeling the relationship between housing prices and the influencing factors, many researchers have studied and quantified the spatial autocorrelation of housing prices using spatial statistics, exploratory spatial data analysis and other methods [9–11]. For example, Marco Helbich et al. used Austrian single-family housing price data to study global and local weighted feature price models [12]. In particular, a mixed geographic weighted regression (MGWR), which avoids the limitations of fixed effects by exploring spatially stable and nonstationary price effects, was proposed. Clapp et al. used an autoregressive process to simulate a citywide housing price index, which was used to forecast one-quarter of individual properties in advance, and used a series of tests to compare the prediction error (PE). The empirical distribution of the PE revealed important information [13]. However, most of these models are simple models, such as linear or logarithmic models, which ignored the non-linear relationship between housing price and related influencing factors. Furthermore, when influencing factors are involved and spatial heterogeneity exists, due to the general existence of spatial autocorrelation, traditional hedonic pricing models might produce biased results.

This paper proposes a framework for modeling and evaluating the spatial distribution of housing prices based on geographic big data. First, variables are constructed to quantify the factors affecting housing prices, such as commercial development, transportation, infrastructure, location, education, environment, and residents' consumption levels. Then, Moran's I tool and geographic detector were used to explore the spatial distribution of housing price data, and the spatial heterogeneity was found. Finally, the XGBoost is used to analyze the important ranking of house price influencing factors, and a zonal nonlinear feature price model based on spatial heterogeneity is constructed. Taking Shenzhen as an example, this paper analyzed the spatial distribution law factors influencing housing prices in Shenzhen and obtains the detailed spatial structure of the housing price market in Shenzhen through local spatial autocorrelation analysis of the model results. Further analysis of abnormal results can provide recommendations for urban renewal and the selection of high-end residential areas by real estate developers.

## 2. Literature Review

### 2.1. The Law of Temporal and Spatial Evolution of Housing Prices

The distribution of housing prices in a city represents a variety of information, such as the development of the city, the distribution of resources, and the spatial distribution of people with different income levels [14,15]. According to William Alonso's ideal house price curve model: the highest house prices appear in the city center, and given that other conditions remain unchanged, house prices decrease monotonically as the distance to the city center increases [8]. This model requires multiple assumptions. When the city has multiple centers or the industrial center deviates from the city center or when high-polluting companies are present in the city center, the spatial distribution of housing prices will not conform to the ideal price curve [16–18].

In fact, the temporal and spatial evolution of housing prices has a high degree of spatial agglomeration and is simultaneously affected by macro factors, such as inflation and population growth [11]. For example, Shao et al. noted in 1997 that accurate estimation of current housing prices in big cities is important for commercial and research investigations in the housing and mortgage markets, and they believe that the spatial structure, especially that of housing prices, has strong spatial agglomeration [19]. Girouard et al. noted that the extent to which housing prices are fairly valued depends largely on whether long-term interest rates are maintained at or close to the current low level; if housing prices are to be adjusted downwards, historical records indicate that the decline may be large and due to low inflation. Moreover,

house prices may continue to fall [20]. Capozza et al. studied the dynamics of actual housing prices using sequence correlation and mean regression coefficients estimated from a panel dataset of 62 urban areas from 1979 to 1995 based on the city size, actual income growth, population growth and actual construction cost, different cross-analyses of serial correlation and regression parameters. In metropolitan areas with high real incomes, high population growth, and high actual construction costs, the serial correlation is higher [21].

### 2.2. Factors Affecting Housing Prices

The factors influencing housing prices are usually analyzed in terms of macroeconomics, policies and laws, market supply and demand, the inherent properties of housing, and location properties [22–24]. A better understanding of the relationship between all of these factors and house prices can provide more complete and relevant information for policymakers in order to improve overall neighborhood quality and create asocial and economic balance between metropolitan areas. On the macro level, Tu et al. studied the dynamic interaction between housing price fluctuations and a country's overall macroeconomic variables. On the one hand, VAR and Granger causality and variance decomposition (VDC) analysis show that house price fluctuations are largely affected by house price increases, house sales growth rates, and population growth rates. On the other hand, volatility affects the GDP growth rate, house price appreciation rate, sales growth rate and volatility itself [25]. Watson et al. studied the relationship between the significant decline in UK housing prices during the period of 2007–2009 and the adjustment of the national economic structure. They found that the structure of housing prices in the UK was centered on a new type of housing ownership model, which was influenced by the Labor government's gradual shift to asset-based welfare. Following the influence of the special concept of financial literacy in the process of system transformation, the welfare recipients of the new model and the housing owners of the new model are jointly constituted in a way consistent with the "House Price Keynesian" new British growth system [26].

At the micro level, Deng et al. studied the relationship between land prices and housing prices. Prior to the adoption of the new method of land sales in 2002, the national, eastern and western housing price index to land price index obeyed a one-way Granger causality relationship but, after 2002, there was a two-way Granger causality relationship between land prices and house prices, which proved the neoclassical land rent theory and Ricardo's rent theory [27]. Luttik et al. studied the impact of environmental factors on housing prices and found that the most important factor for house price increases is the surrounding environment (up to 28%). Moreover, they proved that if the residents of the house are in a good mood, the houses they live in also have relatively high values and high prices [28]. Boyle et al. studied the house price hedonic technique in 2001 and considered air quality, water quality, distance to toxic or potentially toxic locations and the impact of the surrounding environment in the model. If the price is estimated over time and with respect to changes in the role of information in consumer behavior, the above environmental factors are expected to play an important role in the model [29]. Specifically, the influential factors can be classified into location and neighbourhood variables in the hedonic-based model [2,30,31]. According to the 'Law of Rent' theory, the location factor influences the purchasing desire of residents by impacting their transportation expenses, therefore becoming an essential factor influencing the house value [32]. Neighbourhood factors refer to the value of the facilities within a distance of the house to residents, such as natural environment factors, infrastructure factors, and transportation facilities. However, previous studies rely on official statistics and manual survey data, and the quantity of factors affecting housing prices and spatial resolution are restricted. Recently, with the development of information and communication technology (ICT), we can easily obtain multi-source urban big data [33]. Geo-tagged data are typical data, which are distinguished by enormous volumes and frequent update rates, allowing for more thorough insights into understanding the socioeconomic environment associated with housing prices [34].

*2.3. Housing Price Model*

Most previous studies use characteristic price models, such as linear regression models, spatial autocorrelation models and machine learning models, which use various influencing factors related to housing prices. For example, Clapp et al. used an autoregressive process to simulate the citywide housing price index. This model was used to predict one quarter of individual properties in advance, and a series of tests were used to compare the prediction error (PE). The empirical distribution of the PE revealed important information [13]. Marco Helbich et al. studied the global and locally weighted characteristic price model using single family housing price data from Austria and proposed the mixed geographically weighted regression (MGWR). By exploring spatially stationary and nonstationary price effects, the limitation of fixed effects is avoided, and the global model error is shown to be caused by improperly selected fixed effects [12]. Jackson et al. used a multilevel dynamic model to compare the measurement methods in commercial life data. They used the Monte Carlo algorithm to calculate different factor models via different estimates, including the Bayesian method, Bayesian state space method and frequency principal component method [35].

On the other hand, housing price models have been developed using hedonic-based methods and machine learning. A variety of hedonic-based methods have been used to explore the relationship between housing prices and housing characteristic factors. Meese et al. developed hedonic-based regression approaches to evaluate the effect of market fundamentals on housing price dynamics and compared two methods to evaluate the effect of market fundamentals on housing price dynamics [36]. Bin et al. used semiregression to estimate the hedonic price function and compared the performance of predicting price prediction with traditional parameter models. Measurement and prediction of house sales prices has also been performed [37]. Recently, with the development of artificial intelligence technology, a large number of studies have explored the use of machine learning for machine modeling [38,39]. Bin et al. also used an exploratory tree method, which is an important statistical pattern recognition tool. Taking the Singapore resale housing market as an example, they illustrated the professionalism of the technology in terms of the relationship between budget and income accessories, determining important office decisions, and predicting official business [40]. Wang et al. used support vector machines to build a housing price model; they used a particle swarm algorithm to optimize the parameters of the support vector machine, and proposed the PSO-SVM model to predict housing prices. Experimental results show that the proposed PSO-SVM model has good predictive performance [41]. In general, hedonic pricing models treat a property as a composite good with multiple attributes and provide an efficient way to estimate the price of different housing characteristics, including location and environmental factors. Hedonic pricing models typically use ordinary least squares (OLS) for housing price estimates [42], assuming that housing-related factors are independent and identically distributed. However, such methods ignore the non-linear relationship between variables and housing prices, and ignore the spatial autocorrelation of variables.

## 3. Study Area

The research area of this study is Shenzhen, a megacity in China, located on the east side of the Pearl River Delta in northern Hong Kong. Since China implemented the reform and opening policy in 1979, Shenzhen has developed from a coastal village to a large city with a population of 20 million, covering an area of approximately 1996 square kilometers. At the same time, housing prices have undergone tremendous changes. As shown in Figure 1, Shenzhen has 10 administrative regions. The four central districts in the south, Nanshan, Futian, Luohu District, and Yantian, are early special economic zones. A large number of high-tech companies and financial institutions are located in this area, and housing prices are relatively high. The remaining six areas in the northern and eastern suburbs, namely Baoan, Guangming, Longhua, Longgang, Pingshan, and Dapeng, are nonsocial economic zones that include many urban villages, factories, parks, water bodies

and nature reserves. Notably, due to different levels of development, rental prices in the six northern areas of Shenzhen are generally lower than those in the four central areas.

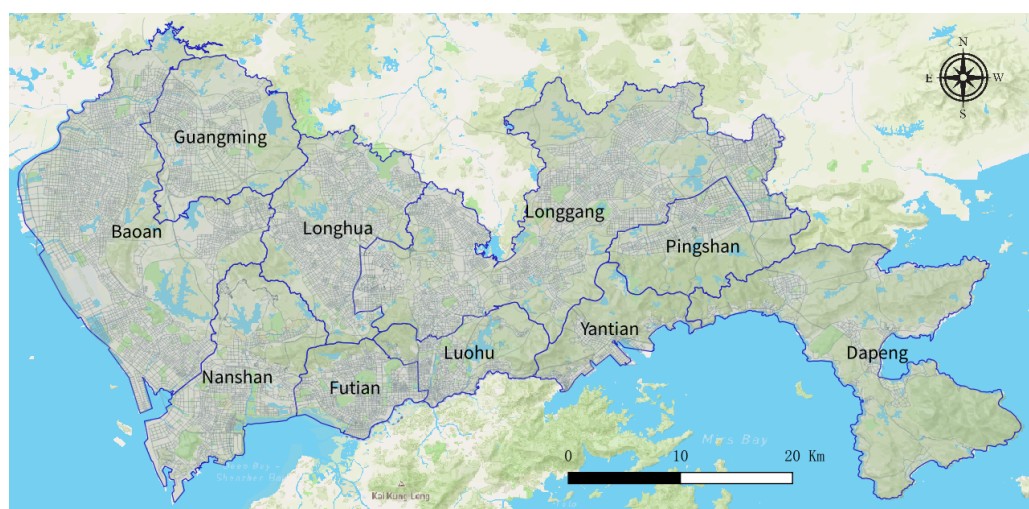

**Figure 1.** Study area: Shenzhen, China.

## 4. Methodology

This paper proposes an analysis framework of urban housing price distribution and influencing factors based on geographic big data and explores the relationship between housing prices and economic and social factors. As shown in Figure 2, the framework consists of four parts: (1) data collection, including online public secondhand housing data, Baidu POI data, Dianping POI data, Landsat8 data, road network data, and education network data. (2) The statistical analysis. Through statistical analysis of original housing price data, spatial autocorrelation analysis and spatial heterogeneity analysis, the statistical characteristics of Shenzhen housing prices are obtained. (3) Associated factors including construction and establishment of relevant indicators. Based on the obtained multisource data, microlevel indicators of factors affecting housing prices, including commercial development, transportation, infrastructure, location, education, environment, and residents' consumption levels, are established. (4) Regression analysis. The spatial statistical characteristics of housing prices and factors that affect housing price are used to construct a zonal nonlinear feature price model. The local Moran index is used to analyze the spatial structure of Shenzhen's housing price market. The detailed steps are as follows.

### *4.1. Data Collection*

#### 4.1.1. House Price Data

This paper used a web crawler to build a data capture program for Shenzhen's second-hand housing price listings based on Fang Tianxia, which is one of the largest real estate information providers in China. We collected housing price data for 4529 neighborhoods in Shenzhen in 2017, including attributes such as the neighborhood's name, price, number of units, latitude and longitude. The final housing price dataset is constructed via data cleaning, coordinate transformation and other steps.

#### 4.1.2. Point of Interest Data

The point of interest (POI) data come from Baidu Maps, which is one of the largest map providers in China. Baidu POIs are points of interest acquired using Baidu Maps as a basic tool. The attributes of each point include basic information such as the full name, longitude, latitude, address, city, contact number, and type. A total of 16.93 million data points were obtained, including 24 categories of POI data, such as catering, education, medical care, and bus stations. In addition, the consumption level and activity level of each POI is obtained through the public comment platform, the number of comments, monthly

average page clicks, number of stars, ratings, per capita consumption, longitude, latitude and other information to supplement the POI data.

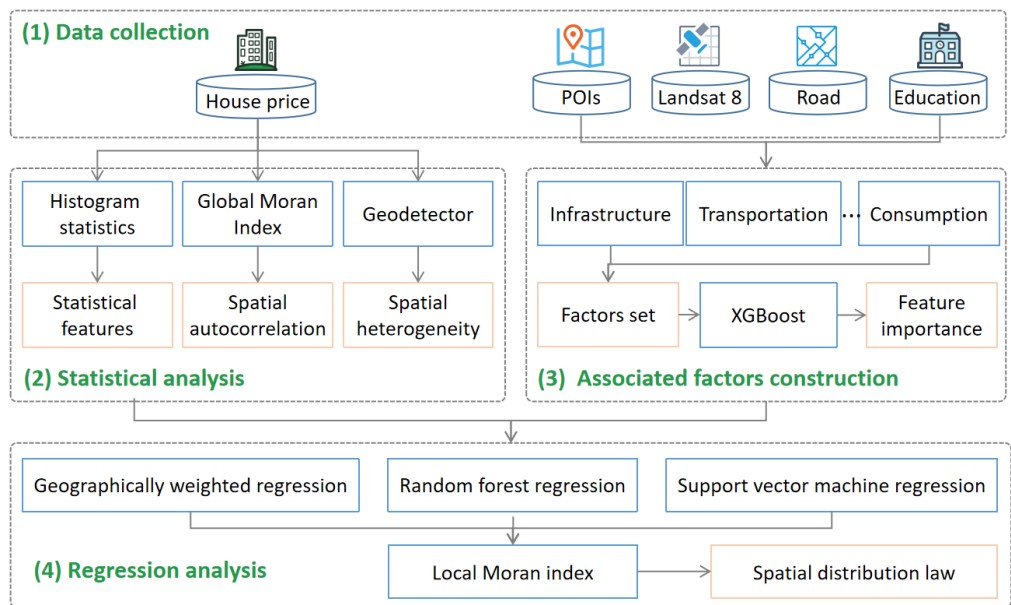

**Figure 2.** Research framework.

### 4.1.3. Remote Sensing Data

Remote sensing images refer to image data formed by satellites receiving the reflection and radiation spectrum information of ground objects through sensors. Remote sensing images contain a wealth of spectral information, texture information, and state information, which can describe the characteristics of the natural environment of the city. This article uses landsat8 images from 27 September 2016 to 27 February 2017, and the spatial range covers the entirety of Shenzhen.

### 4.1.4. Road Network Data

A city's road network structure is also an important part of the city structure. It represents various inherent characteristics of the city, such as the degree of development of the city's traffic, the accessibility of the traffic, and the regional development. In particular, roads of different levels are accompanied by different types and levels of urban functional area structures, which have a multifaceted impact on the surrounding residential prices. This article uses Shenzhen's road network data provided on openstreetmap to describe the road structure of Shenzhen from multiple perspectives. We categorize roads as national roads, provincial roads, highways, urban arterial roads, urban secondary arterial roads, urban branch roads, miscellaneous roads, township roads, county roads and other multilevel roads.

### *4.2. Statistical Analysis*

### 4.2.1. Global Moran Index

The first law of geography states that ground objects with similar locations in space have similar attributes, and spatial correlation analysis can be used to study whether a dataset has agglomeration in space. For example, high values cluster around high values, and low values cluster around low values. Moran's I statistic is often used to measure the degree of spatial autocorrelation, and the calculation is shown in Equation (1).

$$I = \frac{n}{\sum_{i=1}^{n} \sum_{j=1}^{n} w_{i,j}} \frac{\sum_{i=1}^{n} \sum_{j=1}^{n} w_{i,j} z_i z_j}{\sum_{i=1}^{n} z_i^2} \tag{1}$$

where $z_i$ is the difference between the attribute value of sample $i$ and the sample mean, $w_{i,j}$ is the spatial weight between sample $i$ and sample $j$ (e.g., distance reciprocal weight), and $n$ is the number of samples. The range of Moran's I index is $[-1, 1]$. When the value is 0, the house price is not related to the spatial location: house prices present a random distribution in space. The closer the value is to 1, the more the house price is related to location. By contrast, the closer the value is to $-1$, the greater the stepped fault in the distribution of housing prices, with an abnormal situation in the land prices around high-price areas.

4.2.2. Geodetector

This article calculates the degree of difference in housing prices in different administrative regions based on geographic detectors. The core assumption of a geographic detector is the spatial similarity between the independent variables and the dependent variable. Therefore, the dependent variable is partitioned under the condition of discrete independent variables, and the difference between different partitions is the spatial heterogeneity. The $q$ value of the differentiation detector can be used to detect the spatial differentiation of the dependent variable, and it is calculated according to Equation (2).

$$q = 1 - \frac{\sum_{h=1}^{L} N_h \sigma_h^2}{N \sigma^2} \tag{2}$$

where $L$ is the stratification of factor $X$, $N_h$ and $N$ are the numbers of samples in each layer and the whole area, and $\sigma_h^2$ and $\sigma^2$ are the variances of the dependent variable in each area and the whole area. The range of $q$ is $[0, 1]$: the larger the value is, the greater the difference between the dependent variables in each area.

*4.3. Associated Factors Construction*

Due to the complexity of housing market prices, various macro and micro factors impact urban second-hand housing prices from different levels. Based on previous studies on the hedonic-based model [2,30,31], the influential factors can be classified into location and neighbourhood variables. According to the 'Law of Rent' theory, the location factor influences the purchasing desire of residents by impacting their transportation expenses, therefore becoming an essential factor influencing the house value [32]. The distance from the city center can be used as a specific measure for the location factor. Neighbourhood factors refer to the value of the facilities within a distance of the house to residents. The neighbourhood factors include commercial development, transportation, infrastructure, education, environment, and residents' consumption level. The specific factors influencing housing prices are summarized in Table 1. These influencing factors are quantified, and the feature importance model is used to evaluate the degree of influence of different characteristics on urban housing prices.

**Table 1.** Factors affecting house prices.

| Feature Type | Feature Quantification |
|---|---|
| Infrastructure factors | Density of POIs for shopping, food, entertainment, finance, enterprise, life service, and distance to medical POIs |
| Traffic factors | Distance to POIs of transportation facilities, distance to all levels of road network |
| Location factors | Distance to CBD, distance to government agency POIs |
| Educational factors | Average secondary school enrollment rate by housing location |
| Environmental factors | Normalized vegetation index, corrected normalized water index |
| Consumption level factors | Per capita consumption of catering POIs |

### 4.3.1. Associated Factor Indicator

Infrastructure factors: As the foundation of the convenience system, urban infrastructure represents the daily demand of people. With the development of cities, adequate infrastructure has become a necessary component of residential areas. The number of relevant infrastructure POIs within 1500 m of each plot is used to calculate the point density of various POIs to construct infrastructure factor indicators. The evaluation is shown in Equation (3), where $F_1(x, y)$ is the infrastructure factor evaluation index of location $(x, y)$, $Den_i(x, y)$ are the point densities of shopping, catering, entertainment, finance, enterprise, life service, and medical POIs calculated using Baidu POI data, and $i$ is the same for each feature. The coefficient factor used in this article is the random forest feature importance.

$$F_1(x, y) = I_1 Den_1(x, y) + I_2 Den_2(x, y) + I_3 Den_3(x, y) + I_4 Den_4(x, y) + \\ I_5 Den_5(x, y) I_6 Dis_6(x, y) + I_7 Den_7(x, y) \tag{3}$$

Traffic factors: We construct traffic indicators by comprehensively considering the distances of traffic facilities and the distances of road networks at all levels. The calculation is shown in Equation (4), where $F_2(x, y)$ is the evaluation index for traffic factors, $Dis_i(x, y)$ is the distance to various transportation facilities (bus stations, subway stations, etc.), and $Dis_i(x, y)$ is the distance to national highways, provincial highways, highways, urban arterial roads, urban secondary arterial roads, urban branch roads, and miscellaneous distances between roads, township roads and county roads at all levels. Similarly, $i$ is the coefficient of each feature, that is, the feature importance coefficient.

$$F_2(x, y) = I_8 Dis_8(x, y) + \sum_{i=9}^{17} I_i Dis_i(x, y) \tag{4}$$

Location factors: According to the Allson land rent model, we construct a location factor index that considers the distance from the house to the CBD and the distance from a government POI, and the calculation is shown in Equation (5), where $F_3(x, y)$ is the evaluation index of location factors, $Dis_{18}(x, y)$ and $Dis_{19}(x, y)$ are the distance to the Futian Convention and Exhibition Center as the CBD and the density of government POIs, respectively.

$$F_3(x, y) = I_{18} Dis_{18}(x, y) + I_{19} Dis_{19}(x, y) \tag{5}$$

Educational factors: For the measurement of school education resources, this article chooses the school enrollment rate announced by the school as the education factor indicator. Its evaluation equation is Equation (6), where $F_4(x, y)$ is the evaluation index of educational factors, $P_{20}(x, y)$ is the enrollment rate of public junior middle schools in the district where the community is located, and $I_{20}$ is the characteristic importance of the coefficient featured.

$$F_4(x, y) = I_{20} P_{20}(x, y) \tag{6}$$

Environmental factors: This paper obtains the normalized difference vegetation index (NDVI) and modified normalized difference water index (MNDWI) based on Landsat8 data, calculates the average NDVI and MNDWI of each plot, and then obtains the vegetation and water coverage in the community. The calculation formulas of NDVI and MNDWI are shown in Equation (7), where $MIR$ is the mid-infrared band, $NIR$ is the near-infrared band, $R$ is the red band, and $G$ is the green band. $F_5(x, y)$ is the evaluation index of environmental factors, and $I_i$ is the feature importance result of the coefficient factors of each feature.

$$F_5(x, y) = I_{21} \frac{NIR - R}{NIR + R} + I_{22} \frac{G - MIR}{G + MIR} \tag{7}$$

Consumption level: From an economic perspective, housing prices are related to people's income and consumption levels. In this paper, the consumption factor index is constructed through the consumption level of catering POIs. The evaluation equation is

Equation (8), where $F_6(x, y)$ is the evaluation index of consumption level factors, $P_{23}(x, y)$ is the average per capita consumption of restaurants near the community, and $I_{23}$ is the importance of the coefficient factors.

$$F_6(x, y) = I_{23} P_{23}(x, y) \tag{8}$$

### 4.3.2. Factor Importance Analysis

Tree-based machine learning models can be used for feature importance evaluation [43,44]. This paper uses the XGBoost model to analyze the importance of each factor's impact on housing prices. This model has the ability to avoid overfitting and has multiple methods for calculating feature importance, which has been employed in many studies [45,46]. XGBoost is composed of multiple CART regression trees [47]. The final prediction result of the model is the accumulation of the predicted values of multiple decision trees. This approach implements the integrated learning of multiple trees through gradient tree boosting to obtain the final model. The formal definition is in Equations (9) and (10).

$$Obj = \sum_{i=1}^{n} L(y_i, \hat{y}_i) + \sum_{k=1}^{k} \Omega(f_k) \tag{9}$$

$$\Omega(f_k) = \gamma T + \frac{1}{2} \lambda \sum_{j=1}^{T} w_j^2 \tag{10}$$

In Equation (9), $Obj$ is the objective function, where $L$ is the loss function term, that is, the training error, $\hat{y}_i$ is the predicted value, and $y_i$ is the category label of the $i$-th sample. In Equation (10), $\Omega(f_k)$ is the regular term, that is, the sum of the complexity of each tree, the purpose of which is to control the complexity of the model and prevent overfitting, where $f_k$ is the model of the $k$-th tree, $T$ is the number of leaf nodes of each tree, $w$ is the leaf weight value, $\gamma$ is the leaf tree penalty regular term, which has a pruning effect, and $\lambda$ is the leaf weight penalty regular term to prevent overfitting. XGBoost includes several methods for calculating feature importance. The most popular method is calculating the number of times a feature is used to split the data across all trees. Moreover, the average gain of splits which use the feature can be used as feature importance.

### 4.4. Regression Analysis

We developed three zonal nonlinear feature price models considering the spatial heterogeneity and the nonlinear relationship between house prices and housing price influencing factors.

### 4.4.1. Random Forest Regression

There is a nonlinear relationship between housing prices and related factors, and random forest regression can fit this nonlinear relationship well [48]. The random forest model is an ensemble model composed of multiple decision trees. A decision tree is a tree prediction model that inputs the training sample dataset, divides the sample by selecting some features, and selects a feature in each segmentation for threshold division, with the aim of achieving a different final segmentation. The samples of leaf nodes have different category attributes. A random forest is a classifier composed of a series of decision trees $h_k(x)$ $(k = 1, \ldots, N)$, and the marginal function is defined as Equation (11).

$$mg(X, Y) = av_k(I(h_k(X) = Y)) - \max_{j \neq Y}(av_k(I(h_k(X) = j))) \tag{11}$$

where $I(\cdot)$ is an indicator function, $Y$ is the correct classification vector, $j$ is the incorrect classification vector, $av_k(\cdot)$ is the average, and the marginal function represents the degree to which the number of votes obtained in the correct classification exceeds the maximum

number of votes obtained in the incorrect classification. The greater the value is, the greater the confidence of the classifier.

### 4.4.2. Support Vector Machine Regression

There is a complex nonlinear relationship between housing prices and the surrounding built environment that cannot be fitted by simple linear models such as multiple linear regression. Support vector machine regression can deeply dig into the relationship between the built environment and the housing prices of secondhand houses [49]. After model training, the characteristic information such as the density of POIs can be used to calculate housing price information. There is a training sample set $(X, Y) = \{x_{1i}, \dots, x_{si}, y_i\}$, $i = 1, \dots, N$, where $N$ is the number of samples, $X = \{x_{1i}, \dots, x_{si}\}$ is the sample feature set, s is the number of features, and y is the corresponding output value. The basic relationship is shown in Equation (12).

$$Y = \omega^T \varnothing(X) + b \tag{12}$$

where $\varnothing(X)$ is the kernel function, which maps the $X$ vector to a high-dimensional space and makes it have a linear relationship with the output value $Y$, and the $\omega$ and $b$ parameters are solved by the following formula:

$$\min J(\omega, e) = \frac{1}{2}\omega^T\omega + C\sum_{i=1}^{N}(\xi_i + \xi_i^*) \tag{13}$$

$$\text{s.t.} \begin{cases} y_i - \omega^T\varnothing(x_i) - b \le \epsilon + \xi_i \\ \omega^T\varnothing(x_i) + b - y_i \le \epsilon + \xi_i^* \\ \xi_i, \xi_i^* \ge 0 \end{cases} \tag{14}$$

where $\xi_i$ and $\xi_i^*$ are relaxation factors that reduce the noise problem in the data to avoid overfitting. $s$ is the bound of the loss function, that is, a reasonable error between the regression result and the true value is allowed. $C$ is a parameter to achieve a balance between the regression result and the slack variable.

### 4.4.3. Geographically Weighted Regression

The geographically weighted regression model (GWR) introduces the first law of geography to model different regions separately, which can reflect the spatial heterogeneity of housing prices and influencing factors [50]. The greater the spatial heterogeneity of the relationship between housing prices and influencing factors, the more the result is in line with objective reality. Based on the traditional global regression model, the geographically weighted regression model can be expressed as Equation (15).

$$y_i = \beta_0(u_i, v_i) + \sum_{k}^{T} \beta_k(u_i, v_i)x_{ik} + \varepsilon_i \tag{15}$$

where $(u_i, v_i)$ is the geographic location coordinates of the center point of the *i*-th road network segmentation object, $x_{ik}$ is the feature variable of the *i*-th object, $\beta_0(u_i, v_i)$ is the constant sum of other influencing factors considered by the feature variable, $\beta_k(u_i, v_i)$ is the regression coefficient of feature $x_{ik}$, and $\varepsilon_i$ is the random error.

## 5. Result and Analysis

### 5.1. Statistical Law of Housing Prices

Statistical analysis is conducted on the collected secondhand housing data in Shenzhen to understand the distribution of housing prices from a macro perspective. Statistical analysis is performed on the collected secondhand housing prices in Shenzhen communities in April 2018 to obtain statistical characteristics, as shown in Figure 3.

The highest price, in Nanshan, reached 68,196 yuan/square meter, while the lowest price, in Dapeng, was only 33,974 yuan/square meter. As members of the first echelon

of housing prices, Nanshan, Futian and Luohu have concentrated a large number of high-paying enterprises and high-end commercial districts in Shenzhen. The demand for commuting and entertainment has made people more willing to obtain housing close to these areas, causing prices to remain high. The second level of housing prices is in Bao'an and Longhua, both of which are close to the city center, and their high housing prices are also concentrated at the junction with the city center. Other areas such as Longgang and Guangming are far from the city center and hold the basic industries of the city and provide support.

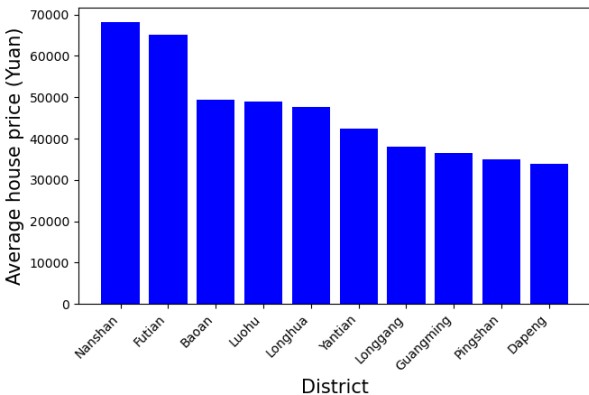

**Figure 3.** Average price of each district.

To test the reasonableness of the housing price data distribution, the housing price data were tested before and after removing the outliers. The QQ chart uses the score of the standard normal distribution as the abscissa and the score of the sample data as the ordinate to draw a scatter plot to illustrate the difference between the data and the standard normal distribution from another perspective. The slope of the straight line in the QQ graph is the standard deviation, and the intercept is the mean. As shown in Figure 4, the housing price data have a normal QQ graph. After log transformation, the scatter points are close to a straight line, that is, the data are transformed to basically conform to the normal distribution.

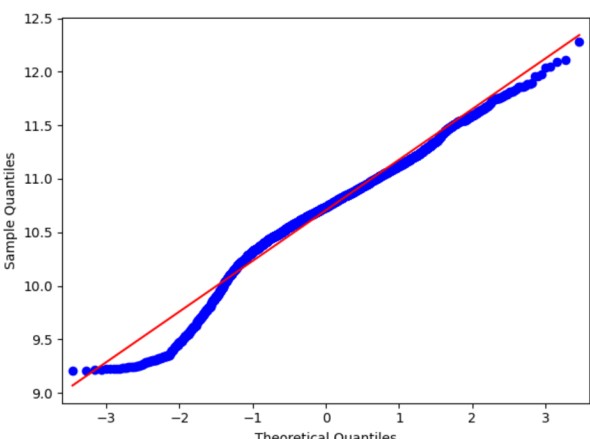

**Figure 4.** Normal QQ graph of house price data.

Statistical analysis shows a big difference in housing prices in Futian, Nanshan and the other eight districts. To measure this difference, this article calculates the degree of difference in housing prices in different administrative regions based on geographic detectors. The statistical Q value obtained by analyzing the housing price data of different administrative regions is 0.534, which provides a measure of the overall differentiation degree of housing prices based on administrative regions, indicating that the distribution

of housing prices in different administrative regions shows a considerable difference. On this basis, the differences among all regions are further analyzed. Under the null hypothesis that there are significant differences in all regions, a *t*-test with a confidence level of 0.05 was conducted, and the results are shown in Table 2, and 'Y' and 'N' denote whether there is a significant difference in house prices between the two regions or not. No significant difference can be observed between Futian district and Luohu District, but there is significant difference between other districts.

**Table 2.** *t*-Test results (confidence level 0.05).

|  | Nanshan | Fotian | Luohu | Yantian | Baoan | Longgang | Pingshan | Guangming | Longhua |
|---|---|---|---|---|---|---|---|---|---|
| Fotian | Y | | | | | | | | |
| Luohu | Y | N | | | | | | | |
| Yantian | Y | Y | Y | | | | | | |
| Baoan | Y | Y | Y | Y | | | | | |
| Longgang | Y | Y | Y | Y | Y | | | | |
| Pingshan | Y | Y | Y | Y | Y | Y | | | |
| Guangming | Y | Y | Y | Y | Y | Y | Y | | |
| Longhua | Y | Y | Y | Y | Y | Y | Y | Y | |
| Dapeng | Y | Y | Y | Y | Y | Y | Y | Y | Y |

Moran's I index was calculated to be 0.327, indicating the spatial aggregation of housing price data. In addition, hypothesis testing is performed, and z-scores (multiple of standard deviation) and *p*-values (probability) are obtained. The z-score is 19.489; that is, no more than 1% probability samples are randomly distributed, which further indicates that valence data are clustered at a high spatial level.

*5.2. The Spatial Distribution and Importance of Associated Factors*

Urban infrastructure, as the basis of the convenience system, is used to satisfy the daily demand of individuals. With the development of cities, adequate infrastructure, such as shopping, catering, entertainment, finance, general hospitals and other related facilities, has become a necessary component of a residential areas. Such infrastructure has improved people's quality of life in many ways and has become a factor considered by people with higher income when purchasing a house. Therefore, the more complete the infrastructure of these communities is, the higher the prices. The distribution of infrastructure indicators in Shenzhen is shown in Figure 5. Store density and catering density are evenly distributed throughout the city, with good store density in most built-up areas. The density of recreational facilities in Futian, Luohu, the Nanshan district and the southern Baoan district is significantly higher than that in other areas. The point density of financial facilities is greatest in Nanshan district, Futian district and Luohu district. The company density is greatest in Nanshan, Futian and Luohu, where high-tech enterprises are located. In addition, there is a high distribution in the north of Bao'an District, where manufacturing enterprises are located in large numbers. In terms of medical resources, regions other than Dapeng have a better allocation.

People with higher income levels have greater purchasing power, so they live in neighborhoods that are relatively more expensive than people with lower incomes. On the other hand, the consumption level of people with high income levels is also high, and the consumption level of restaurants distributed around the community is higher. As shown in Figure 6a, the consumption level index shows a general trend of high consumption in the south and low consumption in the north. The areas with high consumption are concentrated in Nanshan district and Futian District. In addition, Yantian and Dapeng have some tourist attractions and thus higher consumption indices.

Transportation facilities reduce travel costs and expand the reach of an area by improving the convenience of public transportation nearby: housing near subway stations

and bus stations enables individuals to spend less time commuting to school and work. In addition, a city's main roads are the city's traffic arteries, which connect the largest residential areas and work and entertainment areas. Due to the convenience of travel, the nearby housing prices are often higher. The distribution of road networks at all levels reflects the functional distribution of a city, thereby affecting the distribution of housing prices. This paper comprehensively considers the distance to traffic facilities and the road network at all levels to obtain the traffic index factors, as shown in Figure 6b. The overall trend is that the south is stronger than the north and the west is stronger than the east. The traffic index of the central and western regions of each administrative region is higher, and the intensity of the city's construction is highly correlated.

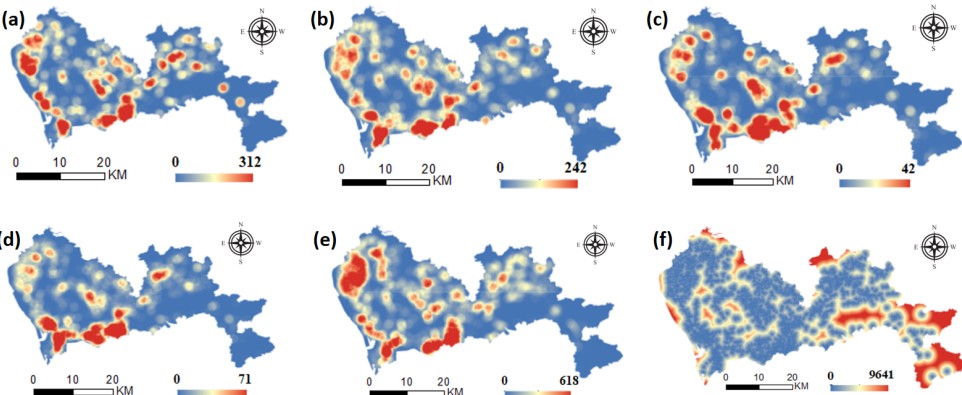

**Figure 5.** Spatial distribution of infrastructure factors. (**a**) shopping density, (**b**) food & Beverages density, (**c**) recreation density, (**d**) finance density, (**e**) enterprises density, (**f**) distance to medical resources.

Residents' expenditures consist of three parts, namely, house purchasing, commuting, and shopping, which are affected by location factors. Therefore, under the condition of fixed expenditure, reducing commuting expenditure is one way to increase house purchase payment. The closer the city center is, the more consumers have the ability to buy high-priced housing. In addition, government agencies are often located in the center of a city, which has good traffic conditions and complete infrastructure and is able to satisfy people's work and entertainment needs. Therefore, the surrounding area is often a local high-value area for housing prices. As shown in Figure 6c,d), we select the central area of Futian as the central area of the city, and government organizations are mainly distributed in Futian and Nanshan.

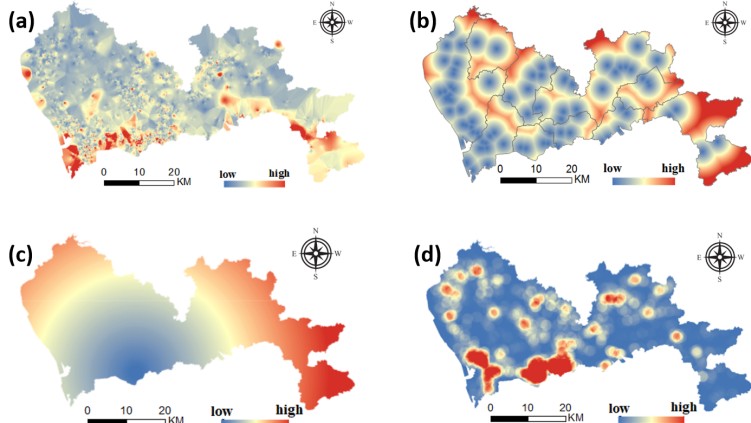

**Figure 6.** Distribution of consumption, transportation and location elements. (**a**) dietary consumption, (**b**) distance to transportation facilities, (**c**) distance to CBD, (**d**) governmental organization density.

The characteristics of POIs reflect the infrastructure construction near the community, but they cannot reflect the natural environment well. The greening rate of a community

is one factor that affects the willingness of high-income people to buy houses. On the basis of remote sensing data, we calculated the vegetation index and water body index to obtain environmental factor indicators, as shown in Figure 7a,b. Generally, there are many mountains and parks in Dapeng, Yantian and Baoan in eastern Shenzhen, and the environmental index is relatively high. In Nanshan district and Futian district, there is also a sporadic distribution of parks, with higher environmental indices.

As the cost of education is increasing, social competition is becoming more fierce, and the imbalance in the spatial distribution of educational resources is becoming more and more obvious. People are increasingly considering educational resources when purchasing houses. This article collects the high school enrollment rates published by schools and on-line education as a quantitative indicator of educational resources. The spatial distribution is shown in Figure 7c. Nanshan has the richest educational resources, followed by Futian, Pingshan and Dapeng.

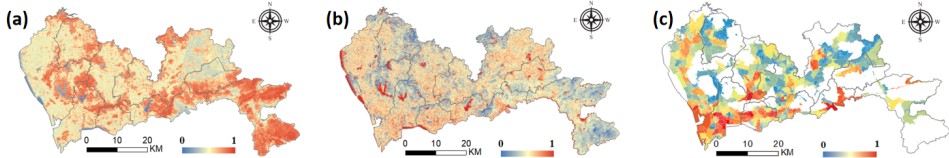

**Figure 7.** Environmental and educational factors. (**a**) vegetation index, (**b**) water index, (**c**) educational resources.

In this paper, the results of feature synthesis are calculated by the XGBoost method. The XGBoost model is sensitive to parameters, where the most important parameters include learning rate and maximum depth of a tree. We set that the learning rate and maximum depth of a tree to be 0.3 and 4, respectively, after a parameter-tuning. The results are shown in Table 3. From a macro perspective, the most important factors are educational factors, regional factors, and infrastructure and environmental factors. Another consideration is to obtain the appropriate activity element. Resources are followed by transportation elements, closeness to transportation facilities and route planning activities for transportation convenience and activity level. These factors explain the development potential of high-income surrounding areas, and the level of consumption can also be based on the existing commercial conditions in the city. Some of the key locations are not suitable for Alonso, a single learning center. There is demand for entertainment, etc., so nearby commercial clusters, greening and infrastructure have become relatively minor factors.

**Table 3.** Feature importance.

| Feature Type | Feature Quantification | Importance Index |
|---|---|---|
| Infrastructure factors | Density of shopping POIs | $3.95 \times 10^{-2}$ |
| | Distance to medical POIs | $3.88 \times 10^{-2}$ |
| | Density of food POIs | $4.08 \times 10^{-2}$ |
| | Density of entertainment POIs | $4.20 \times 10^{-2}$ |
| | Density of financial POIs | $4.16 \times 10^{-2}$ |
| | Density of enterprise POIs | $4.13 \times 10^{-2}$ |
| | Density of life services POIs | $4.15 \times 10^{-2}$ |

**Table 3.** *Cont.*

| | | |
|---|---|---|
| | Distance to transportation facilities POIs | $3.95 \times 10^{-2}$ |
| | Distance to national highway | $4.42 \times 10^{-2}$ |
| | Distance to provincial road | $4.35 \times 10^{-2}$ |
| Transportation factors | Distance to high speed | $4.51 \times 10^{-2}$ |
| | Distance to the main road | $4.75 \times 10^{-2}$ |
| | Distance to secondary arterial road | $4.57 \times 10^{-2}$ |
| | Distance to city branch road | $4.88 \times 10^{-2}$ |
| | Distance to miscellaneous road | $5.51 \times 10^{-2}$ |
| | Distance to country road | $4.85 \times 10^{-2}$ |
| Location factors | Distance to CBD | $5.07 \times 10^{-2}$ |
| | Government POI point density | $4.22 \times 10^{-2}$ |
| Educational factors | Secondary school enrollment rate | $5.58 \times 10^{-2}$ |
| Environmental factors | NDVI | $3.57 \times 10^{-2}$ |
| | MNDWI | $3.96 \times 10^{-2}$ |
| Consumption level | Per capita consumption of catering POIs | $5.08 \times 10^{-2}$ |

Specifically, among the infrastructure factors, the most important is the density and distance to medical facilities. In terms of transportation facilities, the distance from the transportation facility is more important than the distance to the road network. Among location factors, the distance to the CBD is more important than the density of government POIs.

*5.3. Regression Analysis Results*

Taking the house price influencing factor system as the independent variable, the results of secondhand housing prices in Shenzhen were obtained using the random forest regression model, support vector machine regression model, linear regression model and geographically weighted regression model. After evaluating the models, the accuracy of each algorithm is shown in the Table 4.

**Table 4.** House price regression model performance.

| Modle | RMSE | $R^2$ |
|---|---|---|
| SVM_RBF | 10,651.342 | 0.362 |
| SVM_LINEAR | 10,979.052 | 0.346 |
| RFA | 12,888.228 | 0.278 |
| LINEAR | 17,502.941 | 0.296 |
| GWR | 15,748.661 | 0.446 |

Support vector machines have different effects when using different kernel functions. The Gaussian kernel function, which is suitable for nonlinear relations, has an RMSE = 10,651.342 and $R^2$ = 0.362. The linear kernel function, which is suitable for linear relations, has an RMSE = 10,979.052 and $R^2$ = 0.346. Random forest has an RMSE = 12,888.228 and $R^2$ = 0.278, and the contrast linear regression model has an RMSE = 17,502.941 and $R^2$ = 0.296. By comparing the Gaussian kernel SVM and linear kernel SVM, and comparing SVM, RFA and other nonlinear fitting models and linear regression, it can be seen that the relationship between influencing factors and housing prices is more nonlinear than linear. A complicated relationship exists between transportation, education, consumption level, etc. and housing prices, which reflects the necessity of research.

Based on the spatial heterogeneity of housing prices in Shenzhen, a geographically weighted regression model is introduced for comparison. The approach models each spatial object, introduces the results of each object into the model as parameters, and obtains a model that is more in line with the laws of geography. The results are as follows: RMSE = 15,748.661 and $R^2$ = 0.446. The accuracy of this model is higher than the best accuracy achieved by linear regression partition modeling, which indicates there are differences in the relationship between the housing prices of different geographical areas and the influencing factors. However, this independent modeling method cannot compensate for the difference between the linear model and the nonlinear model and even entails a substantial time cost for large sample sets.

The development of Shenzhen is uneven, and differences exist in the degree of transportation convenience, education level, infrastructure construction and housing prices in different administrative regions. The above statistical and spatial heterogeneity analysis indicates that Futian district and Nanshan district are different from each other. Significant differences in the distribution of housing prices are also observed in the other eight districts; that is to say, there may be differences in the law of housing prices in the two parts of the study. Therefore, this article divides the sample to model different regions separately, and the accuracy of the obtained model is shown in Table 5.

**Table 5.** Zonal nonlinear feature price model performance.

| | Nanshan and Futian | | Others | |
|---|---|---|---|---|
| | **RMSE** | $R^2$ | **RMSE** | $R^2$ |
| SVM_RBF | 9433.347 | 0.529 | 7906.373 | 0.582 |
| SVM_LINEAR | 10,997.567 | 0.376 | 8329.878 | 0.504 |
| RFA | 9687.463 | 0.481 | 7580.123 | 0.627 |
| LINEAR | 18,546.651 | 0.212 | 15,974.682 | 0.336 |

For the two areas of Futian and Nanshan, the accuracy obtained by the SVM with a Gaussian kernel is RMSE = 9,433.347 and $R^2$ = 0.529, the result of the SVM with a linear kernel is RMSE = 10,997.567 and $R^2$ = 0.376, the RFA reaches an RMSE = 9687.463 and $R^2$ = 0.481, and the linear regression has an RMSE = 18,546.651 and $R^2$ = 0.212. For the remaining eight areas, the Gaussian kernel SVM has an RMSE = 7906.373 and $R^2$ = 0.582, linear kernel SVM has an RMSE = 8329.878 and $R^2$ = 0.504, RFA has an RMSE = 7580.123 and $R^2$ = 0.627, and the linear regression has an RMSE = 15,974.682 and $R^2$ = 0.336. By modeling the distribution of housing prices, it can be seen that the accuracy is improved when using SVM and RFA to fit the nonlinear relationship. The best accuracy is the result of RFA in areas other than Futian and Nanshan districts: an RMSE of 7580.123 and an $R^2$ of 0.627 are achieved. By comparing the models of the two regions, it can be seen that the accuracy of the models in the other eight districts is generally higher. The average price of secondhand houses in Futian and Nanshan districts is higher than the average price in the other eight districts, and the difference in housing prices is even greater. Moreover, the greater difficulty of learning results in lower model accuracy. In addition, the overall accuracy of linear regression is relatively poor, which illustrates the necessity of using machine learning algorithms for modeling. As shown in Figure 8, the spatial distribution of housing prices obtained by the RFA model and the SVM_LINEAR model in the whole area modeling and zoning modeling is consistent. Overall, second-hand housing prices in Shenzhen are monocentric, with high values concentrated mostly in Futian, Nanshan, and Luohu districts, and steadily declining from Futian as the center.

To further analyze the agglomeration and differentiation of housing prices in Shenzhen, the local Moran's I index, combined with hypothesis testing, is used to analyze the high-value clustering area, low-value clustering area and abnormal area of secondhand housing. The analysis is implemented based on the clustering and contour analysis module under the spatial statistics toolbox in ArcGIS, and the following result diagram is obtained by

analyzing the results of the most accurate regional RFA model. HH indicates clustering of high values near a high value, LL indicates clustering of low values near a low value, HL indicates clustering of low values near a high value, LH indicates clustering of high values near a low value, and Null indicates a random distribution of housing prices.

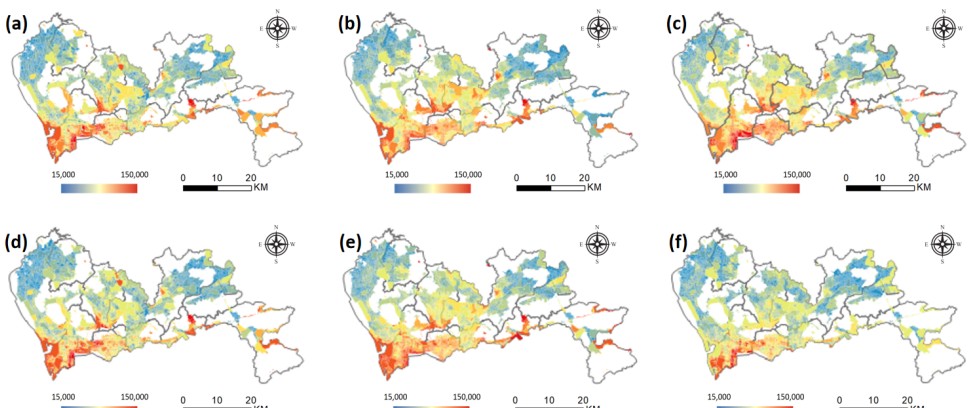

**Figure 8.** Spatial distribution of predicted housing prices. (**a**) RFA model; (**b**) SVM_LINEAR model; (**c**) SVM_RBF model; (**d**) zonal RFA model; (**e**) zonal SVM_LINEAR model; (**f**) zonal SVM_RBF model.

Figure 9 shows that for a significance level of 0.05, Nanshan, Futian, and Yantian show obvious high and high agglomeration, which means that their house values are reflected in the higher part of the overall Shenzhen housing price market, revealing Shenzhen's high-level single-center development model. Relatively low- and medium-priced second-hand houses in the Shenzhen housing market are concentrated in the northwest corner of Shenzhen, that is, in the north of Bao'an district and the west of Guangming new district, and some are in the part of Longgang district near the border of Shenzhen and the northern part of Pingshan new district. Generally, housing prices in suburban residential areas far from the city center are lower, which further illustrates the heterogeneity of the housing price distribution in Shenzhen and the necessity of zoning modeling. In addition, the southern part of Bao'an district, the southern part of Guangming new district, Longhua new district, and the southwestern part of Longgang district are randomly distributed. These areas are located between high-value areas and low-value areas and are also in the urban spatial structure and resource layout. Because of their relatively close proximity to the city center, these areas have become a choice for higher-income groups. Moreover, due to the relatively poor infrastructure and urban resources, the housing prices show strong randomness.

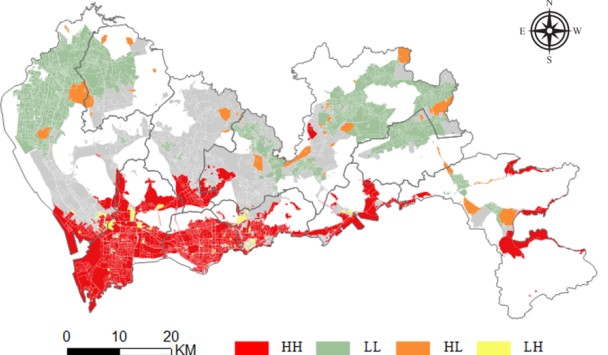

**Figure 9.** Spatial autocorrelation of housing price distribution.

## 6. Conclusions

This paper constructed a housing price analysis framework based on geographic big data. First, based on the hedonic price model and geographic big data, several variables of

housing price influencing factors are constructed. Then, housing price data are used for statistical and spatial analysis to explore the law of spatial distribution of housing prices. Finally, zonal nonlinear feature models are developed to conduct a regression analysis of housing prices and related factors. Taking Shenzhen city as an example, the following conclusions are drawn. (1) Statistical analysis of housing price data in Shenzhen indicates that the average value of housing prices in different districts varies, with large differences between Nanshan district and Futian district and the other eight districts. Moreover, spatial autocorrelation analysis indicates a spatial aggregation phenomenon in Shenzhen housing prices, which has a continuous surface that can be fitted by a function rather than a random distribution. (2) Micro-influencing factors, including infrastructure, transportation, location, education, environment and residents' consumption level, are used to explore the relationship between housing prices and influencing factors. Further, the XGBoost model is used to analyze the degree of influence of each influencing factor on housing prices in the nonlinear model. The order of importance of the housing price influencing factors is: education factors, transportation factors, consumption levels, location factors, infrastructure factors and environmental factors. (3) A zonal nonlinear feature model considering spatial heterogeneity is proposed. Compared with different nonlinear, linear, and geographically weighted regression models, the results show that the zonal nonlinear feature model with better performance to explain the law of the housing price distribution in Shenzhen.

This study still leaves room for future explorations. The influencing factors of housing prices considered in this article are the infrastructure and natural environment around the house, but housing prices are also affected by the community's own structural factors, such as the time the house was built and the structure of the house. Furthermore, the accuracy of modeling environmental factors with medium-resolution landsat8 images does not meet the purpose of refined analysis. Future research can improve the accuracy of the model and strengthen the authenticity of the results by introducing more data, such as house attribute data, and using street view to obtain more detailed environmental conditions.

**Author Contributions:** Conceptualization, X.J.; methodology, X.J. and Z.J.; validation, Z.J., L.L. and T.Z.; data curation, X.J., Z.J., L.L. and T.Z.; writing—original draft preparation, Z.J.; writing—review and editing, X.J., Z.J., L.L. and T.Z.; visualization, Z.J. All authors have read and agreed to the published version of the manuscript.

**Funding:** This research was funded by The Technical Key Project of Shenzhen Science and Technology Innovation Commission under Grant JSGG20201103093401004 and Natural Science Foundation of Guangdong Province under Grant 2019A1515011049.

**Institutional Review Board Statement:** Not applicable.

**Informed Consent Statement:** Not applicable.

**Data Availability Statement:** The data presented in this study are available on request from the corresponding author.

**Conflicts of Interest:** The authors declare no conflict of interest.

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
