# Peer review of "Understanding Housing Prices Using Geographic Big Data: A Case Study in Shenzhen"

_sustainability, doi:10.3390/su14095307_

Round 1
Reviewer 1 Report
Using machine learning methods in predicting spatial-temporal outcomes has been well-developed in the literature. The contribution of the submitted manuscript seems to be limited given the existing work. Some of the statements and assumptions are rather arbitrary without sound justifications. For example the following statement: "The highest house prices appear in the city center, and given that other conditions remain unchanged, house prices decrease monotonically as the distance to the city center increases." Another example is that the authors stated that "As shown in Figure 4, the housing price data have a normal QQ graph." However, due to the significant deviations on the tails, the data are clearly not Normal.
Reviewer 2 Report
Comments: Major
This paper studied the impact factors of housing prices by applying multiple nonlinear models and compared the results of different models. The paper is well-written, but it still has several issues that need to be explained or amended before I can recommend publication.
- In the literature review, the authors need to summarize the research gaps, which correspond to the contributions of this paper, e.g. the necessity and advantage of using non-linear models, the comparison between different nonlinear models, and the reason why regional factors are important for studying housing price, etc.
- The authors need to explain why they use the XGBoost model to get the feature importance. There are many machine learning models that can provide relative importance ranking. For example, GBDT model is used in many researches to obtain relative importance of the features , See Shao et al., 2020. Threshold and moderating effects of land use on metro ridership in Shenzhen: Implications for TOD planning; Zhang et al., 2022. Incorporating polycentric development and neighborhood life-circle planning for reducing driving in Beijing: Nonlinear and threshold analysis. So what are the advantages of XGBoost? Please explain why the XGBoost model was chosen. In addition, the mechanism of XGBoost model is lack.
- The authors need to explain the robustness of the model. XGBoost model is sensitive to parameters and model settings and the feature importance may vary. The authors need to show details of the parameter-tuning process and robustness test in the article.
- (Line 70-72) This may only refer to the assumption of Alonso model. The authors need to clarify.
- (Line 257-260) Based on which papers? Need citation. Besides, a more explicit discussion about the factor selection is also needed (both here and in the literature review).
Round 2
Reviewer 1 Report
I think the authors have addressed most of the comments.
Reviewer 2 Report
The authors have done a good job to respond to comments. The literature review is much stronger. The model selection and the tuning process are more convincing. The vague expressions in the former context are improved. I think this paper can be accepted.
This manuscript is a resubmission of an earlier submission. The following is a list of the peer review reports and author responses from that submission.